# Effect of COVID-19 Pandemic on Medical Students—A Single Center Study

**Maria Poluch** [1] , **Robert Ries** [2] **and Monjur Ahmed** [3,*]

1 Sidney Kimmel Medical College, Thomas Jefferson University, Philadelphia, PA 19107, USA
2 Christiana Care Surgical Residency Program, Newark, DE 19718, USA
3 Division of Gastroenterology, Thomas Jefferson University Hospital, Philadelphia, PA 19107, USA
* Correspondence: monjur.ahmed@jefferson.edu; Tel.: +1-304-633-1354

**Abstract: Background:** The COVID-19 pandemic caused medical education to shift unprecedentedly, leading medical schools to switch to virtual platforms and modify student-patient interactions. On top of educational changes, medical students adapted to their support network, finances, and mental and physical health changes. Objective: To understand the holistic impact of COVID-19 on medical students and medical education and identify how to distribute resources during future educational disruptions in a large medical university in the United States. **Methods:** An anonymous online survey was distributed to medical students at Sidney Kimmel Medical College, Philadelphia, in February 2021. Participants self-reported the impact of the COVID-19 pandemic on their medical education, family life, financial burden, mental health, and physical health. **Results:** 168 out of 1088 students at Sidney Kimmel Medical College completed the survey, with 58% (98/168) of the respondents identifying as female. The class breakdown was as follows: 38% (63/168) first years, 18% (31/168) second years, 21% (36/168) third years, 20% (34/168) fourth years, and 2% (4/168) were considered "other" (including research year, Master's program). A total of 28% of respondents reported developing new mental illness, with second years having the highest incidence at 39%. In total, 42% said the pandemic affected a previous mental health condition. Further, 96% of third and fourth years reported COVID-19 affected their clinical rotations. In total, 68% of first years reported their entrance to medical school was severely affected. Moreover, 13% reported losing a family member due to COVID-19, and 7% reported personal sickness due to COVID-19. Additionally, 16% reported incurring a financial burden due to the pandemic. **Conclusion:** COVID-19 impacted the well-being of students by affecting their mental health and financial burdens. Clinical rotations and medical school entrance were the most problematic aspects. In the future setting of major educational disruptions, this study provides a starting point for where to focus resources, mental health support, financial support, and academic flexibility.

**Keywords:** COVID-19; pandemic; medical education; medical students; virtual medical education





## 1. Introduction

The COVID-19 pandemic resulted in unique challenges across all educational fronts. Medical education was forced to change its ways while providing quality education for future physicians. Social distancing, the most important preventative strategy, caused the need for quick adaptions to the typical medical education model. The United States medical education model traditionally includes:

- In-person clinical experience training.
- In-person lectures.
- In-person small group learning during the first two years of medical education.

The third and fourth years of education are typically spent entirely in the hospital doing clinical rotations with direct patient contact, providing students with skills to make

them more comfortable caring for, communicating, and diagnosing patients. The COVID-19 pandemic completely changed this educational model. The pandemic forced medical education to switch to virtual platforms and eliminate student-patient interaction at the beginning of the pandemic. Doctors, nurses, and physician assistants were considered essential workers and were allowed to continue carrying out their work in person. Menon et al. cited how medical students were not essential workers and that medical institutions were responsible for the well-being of students, supporting the suspension of medical students from clinical rotations [1]. In Apr 2020, Newman et al. wrote a paper with a call to action to change medical education for this pandemic [2]. Jumreornvong et al. wrote about taking advantage of incorporating telemedicine into the medical education curriculum during this time [3]. Rolak et al. documented the impacts and challenges on United States medical students during the pandemic, showing that anatomy, clinical rotations, and the United States Medical Licensing Examination (USMLE), boards exams, were all affected in 2020 [4]. The United States was not alone in this challenge. The effect of the change in medical education expands worldwide, with Alsoufi et al. documenting medical students' attitudes around electronic learning in Libya. They also demonstrated the specific effect of online learning on medical students due to the pandemic, with as many as 13% experiencing health difficulties and 15% experiencing psychological consequences [5]. A study at the University of Cyprus showed that medical students' mental health deteriorated when medical education switched to digital learning, cynicism levels increased, and emotional exhaustion increased [6]. Essangri et al. showed that 65% of medical students surveyed in Morocco had experienced physiological distress because of the COVID-19 outbreak [7]. This study quantifies how the COVID-19 pandemic has holistically impacted medical students' education, financial burden, mental health, and physical health at a single United States institution [8].

## 2. Methods

An anonymous online survey was distributed to all Sidney Kimmel Medical College students at Thomas Jefferson University in Philadelphia, PA in February 2021. The Institutional Review Board (IRB) approved this study. This study was conducted in compliance with the ethical standards of the responsible institution on human subjects as well as with the Helsinki Declaration. Participants were asked to prompt questions about how severely they view their medical education had been affected if they lost their family members to COVID-19, the financial burden, and the mental and physical health effects of COVID-19. Participants self-reported the impact of the COVID-19 pandemic on their medical education, family life, financial responsibility, mental health, and physical health.

## 3. Results

In total, 168 of 1088 students at Sidney Kimmel Medical College completed the survey, with 58% (98/168) of responders identifying as female. The student responders breakdown was as follows: 38% (63/168) first years, 18% (31/168) second years, 21% (36/168) third years, 20% (34/168) fourth years, 2% (4/168) were considered "other" (including research year, Master's program). In total, 28% of respondents reported developing new mental illness, with second years having the highest incidence at 39%. Further, 42% said the pandemic affected a previous mental health condition. Additionally, 96% of third and fourth years reported COVID-19 affected their clinical rotations. A total of 68% of first-years reported their entrance to medical school was severely affected. In total, 13% of students reported losing a family member, and 7% reported personal sickness due to COVID-19 infection. Moreover, 16% reported incurring financial burdens due to the pandemic. The summary of survey results is shown in Table 1 and Figure 1.

**Table 1.** Summary of Survey Results.

| Survey Question | *n* | % |
|---|---|---|
| Have you developed any new physical illness during this pandemic? (*n* = 168) | | |
| Yes | 28 | 17% |
| No | 140 | 83% |
| If you answered "yes" to the previous question, was the illness COVID-19? (*n* = 28) | | |
| Yes | 11 | 39% |
| No | 17 | 61% |
| Have you developed any new mental illness (e.g., anxiety, depression, stress, etc.) during this pandemic? (*n* = 167) | | |
| Yes | 47 | 28% |
| No | 120 | 72% |
| Has this pandemic affected your pre-existing mental illness (e.g., anxiety, depression, OCD, ADHD, etc.)? (*n* = 158) | | |
| Yes | 67 | 42% |
| No | 91 | 58% |
| Has this pandemic affected your study in the medical school? (*n* = 167) | | |
| Yes | 144 | 86% |
| No | 23 | 14% |
| If you answered "yes" to the previous question, how would you grade the severity that the pandemic has affected your study? | | |
| Mild | 47 | 33% |
| Moderate | 74 | 51% |
| Severe | 23 | 16% |
| Has this pandemic affected your clinical rotations in the medical school? (Third and Fourth Years only) (*n* = 70) | | |
| Yes | 67 | 96% |
| No | 3 | 4% |
| How was your entrance experience into the medical school affected by the pandemic as a first-year student? | | |
| Not affected | 0 | 0% |
| Mildly affected | 3 | 5% |
| Moderately affected | 17 | 27% |
| Severely affected | 43 | 68% |
| Have you lost a family member due to COVID-19? (*n* = 167) | | |
| Yes | 22 | 13% |
| No | 145 | 87% |
| Are you going through any financial burden because of the COVID-19 pandemic? (*n* = 167) | | |
| Yes | 26 | 16% |
| No | 141 | 84% |

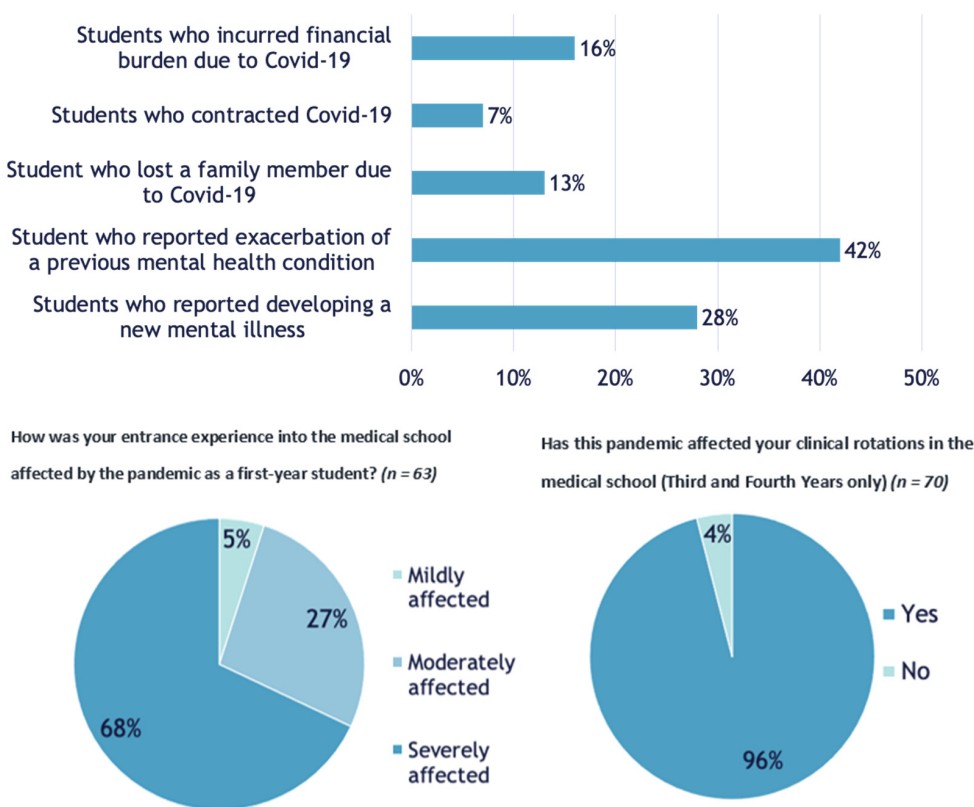

**Figure 1.** Summary of survey results in bar graphs and pie charts.

### 4. Discussion

COVID-19 has affected medical education and population health since March 2020 and called for a rapid adaptation in medical education. It is essential to understand the impact of COVID-19 on medical education and students' well-being, as these students will serve in the future medical workforce. Their experiences during this time will shape their decision as physicians and prime them for the flexibility needed in the medical field. Per the survey results, students at all levels of medical education in 2020–2021 were affected by the pandemic in multiple ways. The most substantial response was developing a new mental illness and exacerbating a previous mental health condition. The first-year students significantly impacted their medical education, as all lectures, including anatomy teaching, were moved to virtual. Second-year students, who were noted to be approaching dedicated study time for their step 1 USMLE examination, which was numerically scored in 2020, had the highest rates of mental illness, although this was not statistically significant. This effect does lead us to consider the stress of step 1 USMLE examination compounded with social isolation at that time. 96% of third and fourth-year students reported their rotations being affected. At the pandemic's beginning, their clinical rotations were entirely on hold. During this time, students participated in virtual rotations. After a few months, they were allowed to do clinical rotations with appropriate precautions. These patient experiences are a critical part of medical education. As vaccines were available, students were permitted in limited clinical environments. Sixteen percent of medical students incurred additional financial burdens. The government responded with loan programs meant to provide relief for students affected by COVID-19. In total, 7% of students reported acquiring COVID-19 infection, and 13% reported losing a family member due to COVID-19. This fact was something many could relate to and added to the distress of students and the community alike. COVID-19 impacted the well-being of students by affecting their mental health and financial burdens. Educational experience, clinical rotations, and entrance to medical school were the aspects most affected. Government loan programs facilitated financial relief, and

the impact of these should be analyzed in the future. The many variations of medical education during this time show the resilience of the medical community. The integration of virtual lectures and simulations should be analyzed for effectiveness. This pandemic forced us to use technology more than we ever have before. It changed the workforce's lifestyle and methods of education. Although not all practices are worth keeping, this allows us to explore further how to support medical school students. The medical school dean decided to protect medical students from the personal risk of catching COVID-19 based on the needs and local pandemic situation. This step was taken as per the guidance of the Association of American Medical Colleges. Similar measures were taken in other medical schools in the United States and worldwide. In future settings of natural disasters or major educational disruptions, this study provides a starting point for where to focus resources—mental health support, financial support, and academic flexibility. Further research should look at programs supporting medical students' mental health and their effects during the pandemic. This study fills the literature gap in documenting the holistic impact of the pandemic on medical students at a single United States institution.

**Author Contributions:** Conceptualization, M.A.; methodology, M.A.; software, R.R.; validation, R.R., M.P. and M.A.; formal analysis; M.P. and R.R.; investigation, R.R., M.P. and M.A.; resources, Sidney Kimmel Medical College, Thomas Jefferson University; data curation, Microsoft Excel; writing—original draft preparation, M.P.; writing—review and editing, M.A.; visualization, M.A. and R.R.; supervision, M.A.; project administration, M.A. Each author contributed equally to complete the study and write the manuscript. All authors have read and agreed to the published version of the manuscript.

**Funding:** This research received no external funding.

**Institutional Review Board Statement:** (IRB Approval, add in the Methods section): IRB approved. Ethical Compliance with Human/Animal Study (add in the Methods section). This study was conducted in compliance with the ethical standards of the responsible institution on human subjects as well as with the Helsinki Declaration.

**Informed Consent Statement:** Voluntary.

**Data Availability Statement:** Available.

**Conflicts of Interest:** The authors declare no conflict of interest.

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
