# Peer review of "Effect of COVID-19 Pandemic on Medical Students—A Single Center Study"

_ime, doi:10.3390/ime1010004_

Round 1

Reviewer 1 Report

- in Introduction: USMLE - explain the abbreviation

- In Methods - Who did create the survey, was it  pilot tested before distributed?

- In Discussion , … 7% of students reported acquiring covid-19, and 13% …. - Repetition of results. I advise to rewrite it.

- In Discussion, ….The medical school dean decided…. - The same decision was taken worldwide. I recommend to discuss it broader not only limited to  your university

Author Response

in Introduction: USMLE - explain the abbreviation: Done.

  • In Methods - Who did create the survey, was it pilot tested before distributed? The authors created the survey. Not pilot tested before.
  • - In Discussion , … 7% of students reported acquiring covid-19, and 13% …. - Repetition of results. I advise to rewrite it: Done.
  • In Discussion, ….The medical school dean decided…. - The same decision was taken worldwide. I recommend to discuss it broader not only limited to  your university: Done
  • English language and style are fine/minor spell check required: Done.
  •  

Reviewer 2 Report

This paper described the impact of the Covid-19 pandemic on 168 medical students regarding their medical education, family life, financial burden, mental health, and physical health. 

Although the paper is single institute research, it highlights how the pandemic holistically influenced their medical education. 

The manuscript is clear, relevant for the field, and presented in a well-structured manner. 

Some minor points should be clarified:

1. The literature gap this paper tries to fill is not clearly described in the introduction. 

2. How the pandemic affected their clinical rotations should be described more clearly.

In summary, this paper can be accepted after minor revision.

Author Response

  1. English language and style are fine/minor spell check required:  Done.
  2. The literature gap this paper tries to fill is not clearly described in the introduction: Now clearly described.

  3. How the pandemic affected their clinical rotations should be described more clearly: Done